# Effects of unilateral and bilateral complex-contrast training on lower limb strength and jump performance in collegiate female volleyball players

**Beiwang Deng**[1☯‡], **Ruixiang Yan**[1☯‡], **Xinke Sui**[1☯‡], **Gesheng Lin**[1], **Shicong Ding**[2*], **Duanying Li**[1,2*], **Jian Sun**[1,2*]

**1** School of Athletic Training, Guangzhou Sport University, Guangzhou, Guangdong, China, **2** Guangdong Provincial Key Laboratory of Human Sports Performance Science, Guangzhou Sport University, Guangzhou, Guangdong, China

☯ These authors contributed equally to this work.
‡ These authors share co-first authors on this work.
* sunjian@gzsport.edu.cn (JS); liduany@gzsport.edu.cn (DL); 11246@gzsport.edu.cn (SD)

## Abstract

The concurrent development of strength and power is considered effective for improving jump performance in athletes.Although previous studies have examined the effects of unilateral and bilateral training, there is a lack of systematic research comparing the impact of these two training modes within the framework of Complex-Contrast Training (CCT) on female volleyball players. This study aimed to compare the effects of Unilateral Complex-Contrast Training (UCCT) and Bilateral Complex-Contrast Training (BCCT) on lower limb strength and jump performance in collegiate female volleyball athletes. A total of 16 female volleyball players were randomly assigned to either the UCCT group (n = 8) or the BCCT group (n = 8) using a random number-based randomization method. The training intervention consisted of 2 sessions per week for 8 weeks.Before and after the intervention, participants underwent a series of standardized tests, including Countermovement Jump (CMJ), Squat Jump (SJ), Drop Jump (DJ), Eccentric Utilization Rate (EUR), Standing Long Jump (SLJ), and one-repetition maximum (1-RM) squat. Statistical analysis was conducted using JASP version 0.18.3.0 with a 2 (group) × 2 (time) two-way repeated measures ANOVA. The results showed significant main effects of time ($p < 0.001$) for CMJ, DJ, SLJ, and 1-RM squat, while SJ and EUR did not exhibit significant improvements. A significant time × group interaction was found for CMJ ($p = 0.009$), and simple effects analysis indicated that the UCCT group showed a more pronounced improvement. No significant main effects of group or time × group interaction effects were observed for the other variables. Overall, UCCT demonstrated similar effects to BCCT in improving horizontal jump performance and maximal lower limb strength, but showed a greater advantage in enhancing vertical jump performance.

---

**Data availability statement:** All relevant data are within the manuscript and its Supporting Information files.

**Funding:** This study was supported by the Guangdong Provincial Philosophy and Social Sciences Regularization Project 2022 (GD22CTY09): Research on the Coordinated Development Path of International Competitiveness in Sports in the Guangdong-Hong Kong-Macao Greater Bay Area.The funders had no role in study design, data collection and analysis, decision to publish, or preparation of the manuscript.

**Competing interests:** The authors have declared that no competing interests exist.

**Abbreviation:** CT, complex training; CCT, complex-contrast training; UCCT, unilateral complex-contrast training; BCCT, bilateral complex-contrast training; CMJ, countermovement jump; SJ, squat jump; DJ, drop jump; EUR, eccentric utilization rate; SLJ, standing long jump; 1-RM squat, one-repetition maximum (1-RM) squat; ICC, interclass correlation coefficients; CV, coefficients of variation

## 1. Introduction

In volleyball competitions, athletes frequently perform high-intensity, explosive movements such as jumping, hitting, and blocking [1,2]. In a high-level match, athletes may execute approximately 250–300 explosive actions, including 50–60% jumping actions, 27–33% attacking movements, and 12–16% ground movements (e.g., digs) [3]. For female volleyball players, in addition to technical skills, tactics, appropriate lean body mass, and higher stature [4,5], jump ability is also one of the key factors for gaining a competitive advantage [6–8]. A higher vertical jump helps reduce the opponent's attack during blocking, and during attacking, it assists athletes in hitting over the net with a better attack angle [9]. Furthermore, quick acceleration and movement during matches are equally crucial. Strong horizontal explosiveness improves athletes' start-up speed when receiving serves and enhances lateral mobility in defense [10].

To improve jumping performance, various training strategies have been widely applied, including resistance training and explosive training. However, existing studies suggest that traditional resistance training alone may not be sufficient to significantly improve the jumping performance of well-trained athletes [11,12]. Newton et al. [13] found that traditional squat training did not effectively enhance the vertical jump height of national-level male volleyball players, whereas ballistic jump squats led to a significant improvement of approximately 6%. Similarly, Häkkinen [14] emphasized the importance of systematically combining high-intensity resistance training with high-speed training to maintain strength and explosiveness throughout the season, in order to prevent the "detraining effect." Even a five-week interruption of heavy-load training followed by only explosive training can result in a significant decline in maximal strength and jumping performance. Furthermore, some studies have shown that traditional slow-speed resistance training can improve vertical jump ability in untrained individuals [15], but the transfer effect is limited in highly trained athletes. Studies also suggest that strength gains without concurrent development of neuromuscular coordination may limit the application of these gains in actual performance [12]. Therefore, the parallel development of strength and explosiveness (rather than separated phases of training) may be more beneficial for improving the actual transfer effect of training adaptations.

In the pursuit of more effective ways to enhance the jumping ability of high-level athletes, researchers have gradually shifted their focus from isolated strength development to integrated training strategies that combine strength and explosiveness [16]. Among these, complex training (CT) has gained widespread attention for its ability to combine high-load resistance training with explosive movements within the same training session, making it a more effective intervention [17]. Existing literature divides complex training into four main types [18]: ① Complex-descending, which involves performing high-load strength training followed by low-load explosive movements, such as completing three sets of 85% 1RM squats followed by three sets of standing long jumps; ② Complex-ascending, which reverses the order by first performing low-load enhancement training and then high-load training, such as performing box jumps followed by high-load squats; ③ Complex-Contrast Training (CCT), where high-load and low-load exercises alternate within

the same training unit, for example, performing a set of high-load squats followed by a set of standing long jumps; ④ French-contrast training, which starts with high-intensity strength training and transitions through moderate-load and various-speed explosive exercises, such as high-load squats → standing long jumps → weighted jumps → resistance band-assisted squat jumps. Among these, the CCT model has received particular attention. In this approach, alternating between high-load and low-load exercises may enhance the force and contraction speed of subsequent low-load exercises through the post-activation potentiation effect phenomenon, thereby improving the effectiveness of subsequent explosive training [18].

Systematic reviews and meta-analyses have shown that compared to other complex training methods, CCT yields higher effect sizes in improving maximal strength (ES = 2.01 vs. 1.29) and vertical jump performance (ES = 0.88 vs. 0.55) [19]. However, most comparisons have lacked a truly volume-and-intensity-matched resistance training group, making it unclear whether CCT's reported benefits stem from potentiation effects or simply from workload differences. For instance, Luders et al. (2024) reported that semiprofessional female Australian Rules footballers achieved similar improvements in 1-RM strength and jump performance when CCT was directly compared with traditional resistance training matched for relative intensity (85% 1-RM) and set–rep structure; the primary advantage of CCT was a reduction in total session time [16]. Similarly, Schneiker et al. (2023) found that subelite male Australian Rules footballers experienced comparable gains in half-squat 1-RM, vertical jump height, and sprint performance when contrast training and progressive resistance training were matched for exercises, volume, and intensity [17]. These findings suggest that, when relative intensity and set–rep structure are equivalent, CCT does not necessarily outperform traditional resistance training in strength or power gains, but may offer a time-efficiency advantage.

As the exploration of different training strategies continues, the mode of execution—whether unilateral or bilateral—has become an important variable influencing training adaptation. Unilateral training refers to training where the load is primarily borne by one leg, such as single-leg squats, Bulgarian split squats, and single-leg jumps; bilateral training refers to exercises where both legs are engaged with relatively balanced load distribution, such as back squats, deadlifts, and double-leg vertical jumps [20]. Based on this, the current study further refines the training modes within the CCT framework into Unilateral Complex-Contrast Training (UCCT) and Bilateral Complex-Contrast Training (BCCT), aiming to enhance the jumping ability of female volleyball athletes through more targeted interventions. Traditionally, bilateral training has been considered the primary method for enhancing strength and explosiveness due to its higher total load stimulus and favorable neural activation effects, and it has been widely applied in competitive sports [21,22]. However, unilateral training has also gained increasing recognition for its functional benefits. It enables more precise activation of specific muscle groups, thereby enhancing localized control and neuromuscular coordinatio [23]. Numerous studies have demonstrated that unilateral training offers significant advantages in improving lower limb strength, jumping ability, sprint speed, and balance [24,25], with Núñez et al. [26] confirming its effectiveness in enhancing muscle mass and functional performance. However, some studies support the superiority of bilateral training in improving double-leg jumping ability and maximal strength [27]. Additionally, other research has pointed out that there are no significant differences between unilateral and bilateral training in improving athletic performance [28].

While extensive research has explored the effects of complex training as well as unilateral and bilateral training individually, no studies have systematically compared the training effects of CCT with different execution modes (unilateral vs. bilateral) for female athletes. In women's volleyball, actions such as one-legged jumps, and short-duration high-intensity movements are commonly seen. These specific characteristics of the sport rely heavily on unilateral force production and neuromuscular coordination. According to the principle of training specificity, resistance training should aim to replicate the biomechanical features and movement patterns of the target sport [29]. Therefore, exploring the intervention effects of different CCT execution methods will help design more targeted training programs. Based on this, the aim of the current study is to compare the effects of UCCT and BCCT on lower limb maximal strength and jumping ability in female volleyball athletes at the university level.

## 2. Methods

### 2.1. Experimental approach to the problem

The study adopts a randomized parallel control trial design, aiming to analyze the effects of UCCT and BCCT on lower limb strength and jump performance in collegiate female volleyball players, providing scientific evidence for training and optimizing jump ability.The study was conducted during the early season phase, with a total duration of 10 weeks, specifically including 1 week of testing procedures and pre-tests, 8 weeks of training intervention, and 1 week of post-testing. Participants underwent twice-weekly training sessions from October 14/10/2024–22/12/2024, with a training interval of 48–72 hours, totaling 16 training sessions. Prior to the intervention, The participants' typical weekly training schedule included three volleyball-specific practice sessions, two conditioning sessions—which incorporated exercises such as bilateral squats and Bulgarian split squats depending on the training cycle—and one competitive match. Therefore, all participants were familiar with the technical execution of both bilateral and unilateral lower limb exercises prior to the start of the study.This routine was maintained throughout the 8-week intervention period to ensure ecological validity. The details of the training programs are presented in Table 1.

After familiarizing themselves with the test procedures and completing pre-testing, 18 participants were randomly assigned to either the UCCT group (n = 9) or the BCCT group (n = 9) using a random number-based randomization method in IBM SPSS Statistics 26. UCCT group performing a complex contrast training program consisting of Bulgarian split squats and unilateral plyometric exercises, while the BCCT group completed a complex contrast training program consisting of squats and bilateral plyometric exercises. Before and after the training intervention, participants underwent tests for one-repetition maximum (1-RM) squat, Countermovement Jump (CMJ), Squat Jump (SJ), Drop Jump (DJ), Eccentric Utilization Rate (EUR), and Standing Long Jump (SLJ). Furthermore, to accurately assess the independent effects of strength training, participants refrained from any other form of resistance training during the intervention period, aside from the assigned training program.

### 2.2. Participants

The sample size was determined using G*Power 3.1.9.7 [30] through a two-way repeated measures analysis of variance (ANOVA). The input parameters were as follows: statistical test type = ANOVA: repeated measures, within-between interaction; statistical power = 0.95; number of groups = 2; number of measurements = 2; and epsilon for sphericity correction ($\varepsilon$) = 1. Based on previous research on an 8-week explosive training program for young male football players and its effect on neuromuscular performance (effect size f = 0.5) [31], the estimated effect size was used. Using these parameters, the preliminary sample size calculation indicated 16 participants. Considering potential sample attrition, 18 female college volleyball athletes were randomly selected from a pool of athletes who met the inclusion criteria.. Inclusion criteria were: (1) at least 2 years of volleyball-specific training and experience in both unilateral and bilateral resistance training; (2) participation in regional-level volleyball competitions and at least one provincial-level or higher university volleyball league; (3) participation in at least two volleyball-specific training sessions per week; (4) no history of lower limb orthopedic injuries, other lower limb conditions, or cardiovascular diseases in the 6 months prior to the experiment. All participants were members of the same university volleyball team and regularly competed in regional-level tournaments, including the

**Table 1. Weekly training program.**

|  | Mon | Tue | Wed | Thu | Fri | Sat | Sun |
|---|---|---|---|---|---|---|---|
| AM | / | Technical | / | Tactical | Training situations (1-on-1) | Match/ Scrimmage | Rest |
| PM | CCT Intervention (UCCT/BCCT) | / | CCT Intervention (UCCT/BCCT) | / | / | Rest or Active Recovery | / |

Note: am: 9:00–11:30; pm: 15:00–17:30.

B subgroup of the Elite Division in the Guangdong Collegiate Volleyball League. They were also selected to participate in the 12th Guangdong University Games, a provincial competition held every four years, representing a relatively high level of athletic performance among Chinese collegiate athletes.

During the intervention, one participant from the UCCT group dropped out due to injury (non-study related), and one participant from the BCCT group was excluded because of an emergency personal matter preventing timely attendance at post-testing. The remaining 16 participants completed all tests and training (Fig 1). No statistically significant differences in height, age, body mass, or training experience were found between the two groups (Table 2). The training adherence rate

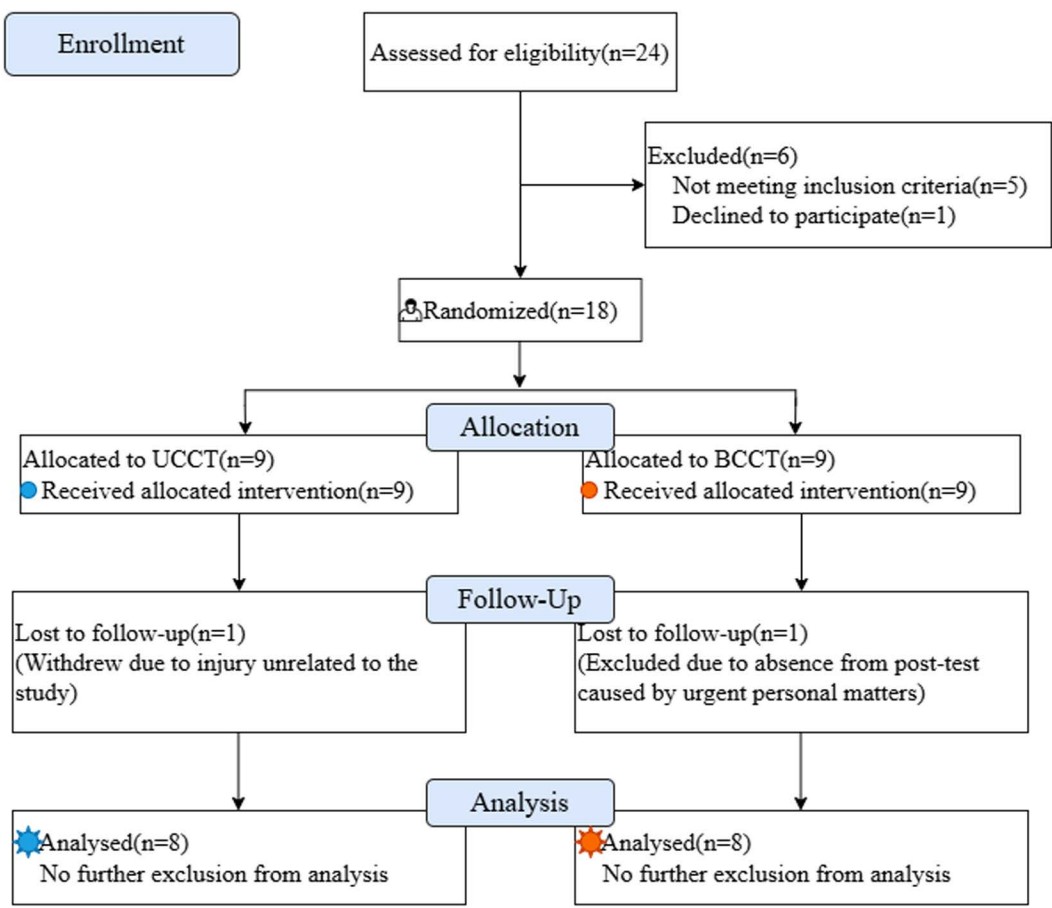

**Fig 1. CONSORT flow diagram.**

**Table 2. Subject information.**

| Basic Information | Group | |
|---|---|---|
| | UCCT (n=8) | BCCT (n=8) |
| Age (years) | 18.75±0.707 | 19.25±0.707 |
| Height (m) | 1.719±0.043 | 1.736±0.056 |
| Body mass (kg) | 63.381±7.142 | 66.879±7.588 |
| Training Years (years) | 5.313±2.712 | 6.375±2.326 |

was 94% (range: 87.5%–100%) for the UCCT group and 93% (range: 87.5%–100%) for the BCCT group. All participants voluntarily participated in the study and signed informed consent forms. The study was approved by the Institutional Ethics Committee (2024LCLL-96) and registered with the Chinese Clinical Trial Registry (ChiCTR2400091000).

### 2.3. Experimental procedures

**2.3.1. Training intervention.** Participants completed an 8-week training intervention, with sessions held twice a week (Wednesdays and Fridays), as detailed in Table 3. Prior to the intervention, all participants underwent a 1-week familiarization training program consisting of 3 sessions, during which they became familiar with the experimental procedures, intervention protocols, testing measures, and the use of relevant equipment. Each training session began with a standardized warm-up lasting 15–20 minutes, including foam rolling, light jogging, dynamic stretching, activation exercises, and movement integration. After the warm-up, participants performed one set of 8 repetitions of barbell squats or Bulgarian split squats with a 20 kg load, followed by one set of 5 repetitions with a 30 kg load. Afterward, participants engaged in unilateral complex contrast training or bilateral complex contrast training according to their assigned group. All training sessions were supervised by two Certified Strength and Conditioning Specialists (CSCS), who ensured proper exercise technique, protocol adherence, and training safety throughout the intervention.

In the UCCT group, participants performed unilateral resistance training with a load equivalent to 85% of their Bulgarian split 1-RM squat, completing 5 sets of 5 repetitions per side, followed by unilateral plyometric training using bodyweight, completing 5 sets of 10 repetitions per side. In the BCCT group, participants completed bilateral resistance training using 85% of their 1-RM squat, performing 5 sets of 5 repetitions, followed by bilateral plyometric exercises using bodyweight, with 5 sets of 10 repetitions. Rest intervals were 3 minutes between sets and 4 minutes between the resistance and plyometric components. Both groups followed an identical training frequency, intensity, and set–repetition structure to ensure consistency in program implementation. The detailed training program is presented in Table 3. After the session, participants underwent foam rolling and static stretching to promote muscle recovery and ensure adequate rest. The plyometric exercises in both groups were selected to incorporate vertical and horizontal movement patterns, aiming to enhance volleyball-specific jump performance by facilitating transfer across directions and reflecting the multidirectional demands of the sport.

**2.3.2. Testing.** Before and after the training intervention, participants underwent lower limb strength and explosive power tests. Each test was conducted on two separate testing days, with at least 72 hours between tests. To ensure consistency and validity of the results, participants were instructed to avoid any vigorous exercise, alcohol consumption, or caffeine intake within 24 hours before the test. To minimize the impact of biological rhythms on the results, all pre- and post-tests were conducted at the same time of day and location, with the same test personnel administering each test to ensure consistency.

**Table 3. Formal training experimental plan for UCCT group and BCCT group [32].**

| Group | Time | Number of sets | Resistance training | | Rest intervals (min) | Plyometric training | | Inter-set rest periods (min) |
| | | | Action | Repeat times | | Action | Repeat times | |
|---|---|---|---|---|---|---|---|---|
| UCCT | Wednesday | 5 | Bulgarian Split Squat | 5/each side | 4 | Single-Leg Hurdle Hop, Rear Foot Elevated Split Jump | 10/each side | 3 |
| | Friday | 5 | Bulgarian Split Squat | 5/each side | 4 | Single-Leg Box Jump, Split Squat Box Jump | 10/each side | 3 |
| BCCT | Wednesday | 5 | Back Squat | 5 | 4 | Double-Leg Hurdle Hop, Squat Jump | 10 | 3 |
| | Friday | 5 | Back Squat | 5 | 4 | Box Jump, Depth Jump | 10 | 3 |

Vertical Jump Tests: The Smart Jump system (Fusion Sport, Australia) was used to assess the Countermovement Jump (CMJ) [33], Squat Jump (SJ) [34], and Drop Jump (DJ) [35]. Participants wore standard athletic clothing and shoes, standing with both feet on the jump mat, waiting for the green light to appear.CMJ: Participants placed their hands on their hips, quickly squatted down, and immediately jumped, keeping their hands on their hips.DJ: From a 30 cm height, participants stepped off the platform, landed with both feet simultaneously, and jumped vertically as quickly as possible.SJ: Participants squatted and held the position for 2–3 seconds before jumping vertically as soon as they heard the signal.In all tests, participants were required to maintain an upright posture in the air, land with both feet simultaneously while bending their knees to absorb the impact. Jump height was recorded in centimeters. Each participant completed 2 trials with 1-minute rest in between, and the best score was recorded to two decimal places.In all tests, participants were required to maintain an upright posture during flight and land with both feet simultaneously, flexing their knees to absorb impact. Jump height was recorded in centimeters. Each participant performed two valid trials with a 1-minute rest between attempts, and the best of the two was used for analysis, recorded to two decimal places.Prior to testing, participants were given standardized verbal instructions emphasizing maximal effort and correct technique. Consistent verbal encouragement ("Jump as high as you can!") was provided during each attempt to promote maximal performance while ensuring all participants received identical motivational input. Trials were repeated if participants failed to maintain upright posture, missed the mat, landed asymmetrically, or failed to land with both feet simultaneously.

Eccentric Utilization Rate (EUR): EUR was calculated using the internationally recognized formula: EUR = CMJ/SJ. The best results from CMJ and SJ were used for this calculation [36]. EUR is commonly used to assess an athlete's ability to utilize the stretch-shortening cycle (SSC) during explosive movements. A higher EUR value indicates more effective use of stored elastic energy and neuromuscular potentiation in the eccentric phase to enhance concentric output, which is particularly important for improving jump performance and lower-limb explosiveness.

Standing Long Jump (SLJ): The SLJ test was conducted on a non-slip surface using a tape measure. A straight line was drawn on the floor to mark the starting point. Participants were instructed to stand behind the line with their toes positioned just behind but not touching or crossing the line. No preparatory movements, such as stepping or hopping, were allowed.Standardized verbal instructions were given before each attempt: "Jump as far forward as you can using both legs, land with control, and avoid crossing the start line before take-off." Participants were allowed to use arm swing during the jump. The distance from the start line to the rear heel at landing was measured. Each participant completed two valid trials with a one-minute rest in between, and the best result was recorded to two decimal places [37].

One-Repetition Maximum Squat: The testing methods for both the Back Squat and the Bulgarian Split Squat followed the guidelines from the NSCA Testing and Evaluation Manual [38]. Participants were asked to avoid any resistance training, running, or vigorous activity 48 hours prior to the test. The 1-RM squat testing procedure is as follows: First, participants perform 10 warm-up repetitions using an empty barbell, followed by a 2–3 minute rest. Then, the weight is increased to 50% of the estimated 1-RM squat, and participants perform 5–10 repetitions, with 3–5 minutes of rest between sets. When the weight approaches 90% of the estimated 1-RM squat, it is increased by 5% at a time, and participants attempt 1–2 repetitions. If the attempt is successful, a 5-minute rest is followed by another 5% increase in weight. If the attempt fails, participants rest for 5 minutes and then attempt again or reduce the weight by 2.5%−5%. The goal is to determine the 1-RM squat within five attempts. In the Bulgarian Split Squat (BSS), the same one-repetition maximum testing protocol as the back squat was adopted. During the test, the dorsum of the non-dominant (rear) foot was placed on a support box of appropriate height, ensuring that the dominant (front) leg performed the movement independently without assistance.To ensure proper form, the knee angle during the back squat and the Bulgarian split squat must reach 90 degrees, meaning the thighs must be parallel to the ground. Three certified spotters ensure safety during the test. The final 1-RM value for both the back squat and the Bulgarian split squat is recorded in kilograms (kg).

The interclass correlation coefficients (ICC) and coefficients of variation (CV) for all test measures demonstrated good reliability, as detailed in Table 4.

**Table 4. Reliability and variability analysis of test indicators.**

| Test indicators | ICC (95%CI) | CV (%) |
|---|---|---|
| CMJ | 0.907 (95%CI: 0.767–0.965) | 9.6% |
| SJ | 0.891 (95%CI: 0.731–0.959) | 9.00% |
| DJ | 0.876 (95%CI: 0.697–0.953) | 6.00% |
| SLJ | 0.856 (95%CI: 0.654–0.945) | 2.89% |

## 2.4. Statistical analyses

Data analysis was performed using JASP version 0.18.3.0, with all data presented as Mean±SD. Group differences in all variable characteristics before the intervention were assessed using independent samples t-tests. Normality of the data was tested using the Shapiro-Wilk test, and homogeneity of variances was verified using Levene's test. If normality was not met, the Wilcoxon signed-rank test was conducted, with results expressed in medians and ranges. The test-retest reliability was evaluated using the coefficient of variation (CV) and the intraclass correlation coefficient (ICC) with a 95% confidence interval. ICCs were calculated in JASP using a two-way mixed-effects model [ICC(3,1)], based on absolute agreement across two baseline trials, and CVs were computed as the ratio of the standard deviation to the mean from the same two baseline trials, expressed as a percentage [39]. ICC values were interpreted for relative reliability [40]: values between 0.5 and 0.75 indicated moderate reliability, values between 0.75 and 0.9 indicated good reliability, and values over 0.90 indicated excellent reliability [41]. Previous reliability studies reported that biomechanical variables with a CV around 10% are reliable, hence a CV ≤ 10% was set as the standard for declaring variable reliability [42,43].

For normally distributed data, independent samples t-tests were used to compare baseline differences in participant metrics. A 2 (group: experimental, control) × 2 (time: pre-intervention, post-intervention) mixed-model ANOVA was conducted to assess the presence of main effects for group and time, as well as the interaction effect between time and group. If a significant time × group interaction was found, simple effects analysis was further conducted, recording F values, p values, and effect size eta squared ($\eta^2$). Eta squared ($\eta^2$) was used to measure the size of effect, with thresholds set as small ($0.01 \leq \eta^2 < 0.06$), medium ($0.06 \leq \eta^2 < 0.14$), and large ($\eta^2 \geq 0.14$) [44].

## 3. Results

All indicators of the subjects followed a normal distribution, with equal variances and no significant differences in pre-test results ($p > 0.05$).

### 3.1. Vertical jump performance

As shown in Table 5 and Fig 2, after 8 weeks of training intervention, the vertical jump ability indicators in the UCCT group and BCCT group showed different results. Specifically, CMJ ($p < 0.001$, $\eta^2_p = 0.729$) and DJ ($p < 0.001$, $\eta^2_p = 0.709$) both showed significant main effects of time, indicating improvements in vertical jump performance over time. while SJ ($p = 0.232$, $\eta^2 p = 0.101$) and EUR ($p = 0.454$, $\eta^2_p = 0.041$) did not show significant main effects of time. CMJ ($p = 0.009$, $\eta^2_p = 0.392$) showed a significant time × group interaction effect. Further simple effect analysis showed that the simple effect of UCCT on CMJ ($p < 0.001$) was significant, while the simple effect of BCCT was not significant effect, indicating that the UCCT group had a greater improvement in CMJ compared to the BCCT group. The time × group interaction effects and main effect of group for other indicators (SJ,DJ,EUR) were not significant.

### 3.2. horizontal jump performance

The SLJ indicator in both the UCCT group and BCCT group showed a significant main effect of time ($p < 0.001$, $\eta^2_p = 0.900$). However, the main effect of group and the time × group interaction effect for the SLJ indicator were not significant for both groups (Table 5 and Fig 3).

**Table 5. Changes in scores between UCT group and BCT group before and after intervention.**

| Index | UCCT (n=8) | | BCCT (n=8) | | Main effect of group | | | Main effect of time | | | Time×Group interaction effect | | |
|---|---|---|---|---|---|---|---|---|---|---|---|---|---|
| | Pre-test | Post-test | Pre-test | Post-test | F | p | η2p | F | p | η2p | F | p | η2p |
| CMJ (cm) | 33.955±3.685 | 36.706±3.647** | 34.044±2.765 | 34.987±2.914 | 0.256 | 0.621 | 0.018 | 37.735 | <0.001 | 0.729 | 9.030 | 0.009 | 0.392 |
| SJ (cm) | 32.602±2.977 | 33.73±3.389 | 33.072±3.118 | 33.496±2.871 | 0.007 | 0.935 | $4.957 \times 10^{-4}$ | 1.564 | 0.232 | 0.101 | 0.322 | 0.579 | 0.022 |
| DJ (cm) | 32.679±1.995 | 36.084±2.377** | 34.196±2.523 | 31.969±1.951* | 1.678 | 0.216 | 0.107 | 34.133 | <0.001 | 0.709 | 1.492 | 0.242 | 0.096 |
| EUR | 1.040±0.038 | 1.073±0.049 | 1.034±0.044 | 1.020±0.065 | 1.791 | 0.202 | 0.113 | 0.594 | 0.454 | 0.041 | 3.615 | 0.078 | 0.205 |
| SLJ (m) | 1.998±0.059 | 2.124±0.072** | 2.025±0.058 | 2.135±0.069** | 0.400 | 0.537 | 0.028 | 126.095 | <0.001 | 0.900 | 0.597 | 0.4553 | 0.041 |
| 1-RM squat (kg) | 77.688±11.171 | 102.438±10.574** | 76.313±10.777 | 97.625±12.386** | 0.342 | 0.568 | 0.024 | 146.153 | <0.001 | 0.913 | 0.814 | 0.382 | 0.055 |

Note: UCCT: Unilateral Complex-Contrast Training; BCCT: Bilateral Complex-Contrast Training;CMJ: Countermovement Jump; SJ: Squat Jump; EUR: Eccentric Utilization Rate;DJ: Drop Jump; SLJ: Standing Long Jump; *P<0.05,**P<0.01.

### 3.3. Maximum strength

Both the UCCT group and BCCT group showed a significant main effect of time for the maximum strength indicator, 1-RM squat ($p < 0.001$, $\eta^2_p = 0.911$). However, the main effect of group and the time × group interaction effect for 1-RM squat were not significant (Table 5 and Fig 4). Both groups showed improvement in maximum strength, but the difference between the groups was not statistically significant.

## 4. Discussion

This study is the first to examine the comparative effects of UCCT and BCCT on maximal lower limb strength and jump performance in female athletes. Following an eight-week intervention, both groups exhibited significant improvements in 1-RM squat, with no statistically significant differences between training modalities. Similarly, DJ and SLJ performance

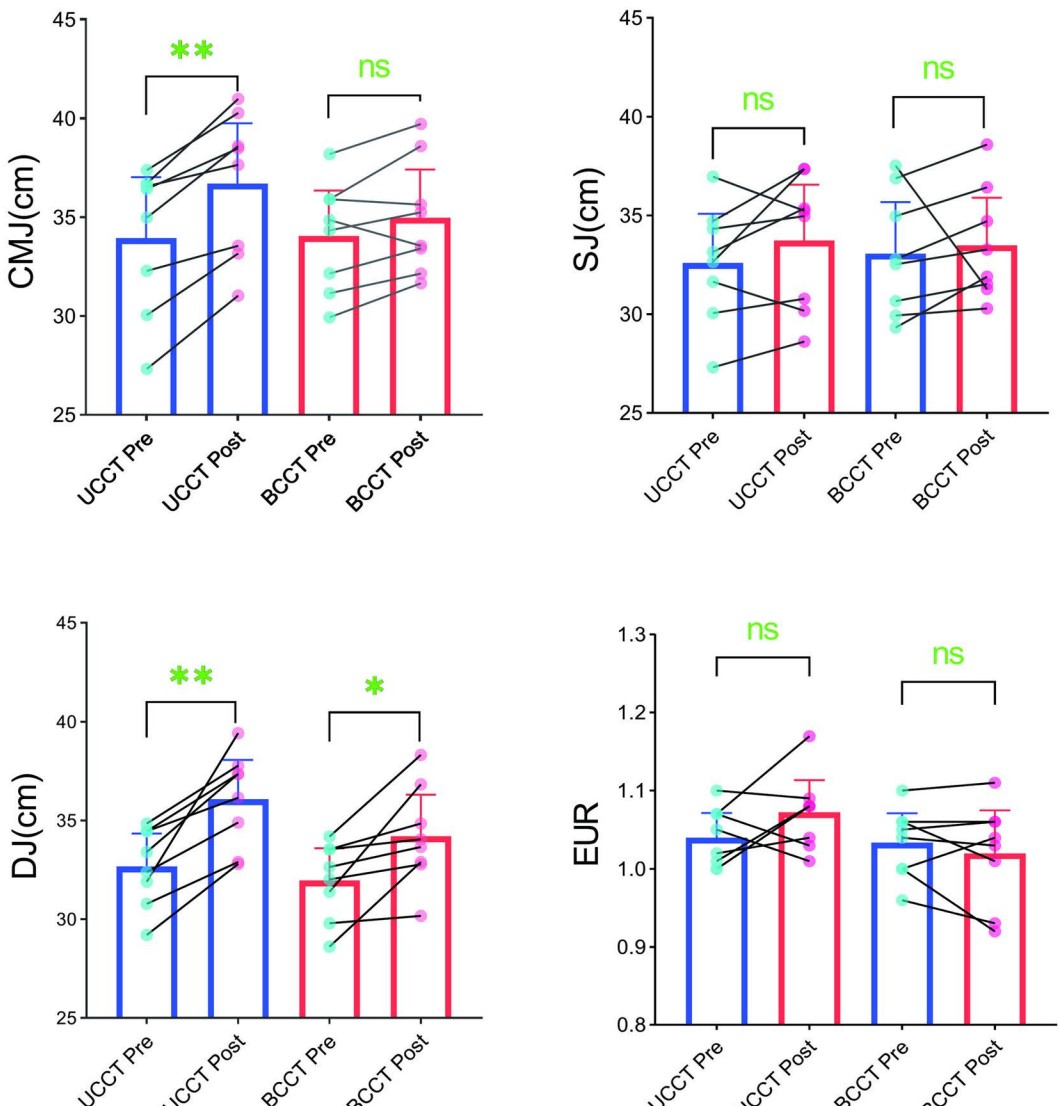

**Fig 2. Individual changes in vertical jump performance following training intervention.** Note: Mean ± 95% CI; *P < 0.05; **P < 0.01.

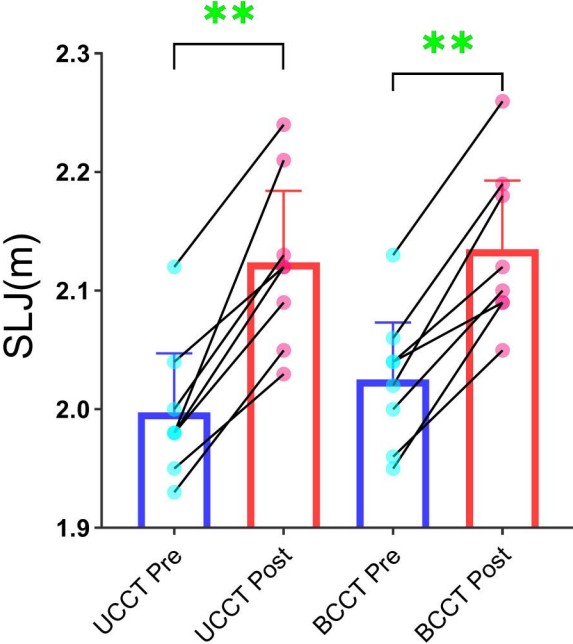

**Fig 3. Individual changes in SLJ following training intervention.** Note: Mean±95% CI; *P<0.05; **P<0.01.

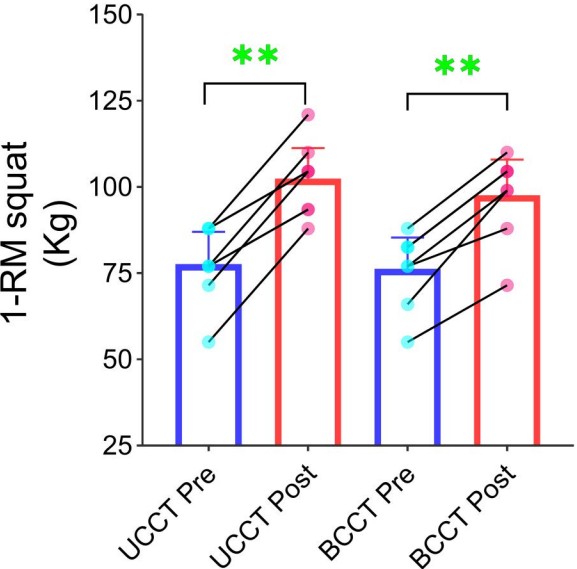

**Fig 4. Individual changes in 1-RM squat following training intervention.** Note: Mean±95% CI; *P<0.05; **P<0.01.All participants completed both assessments; overlapping points indicate identical pre- and post-test values.

improved markedly in both groups. However, CMJ performance increased significantly only in the UCCT group and surpassed that of the BCCT group. These findings imply that both training strategies effectively transfer strength and power gains to bilateral jump tasks, with UCCT offering a possible edge in specific explosive movements such as CMJ and DJ.

These results align with previous research. For instance, Appleby et al. [45] reported that eight weeks of both unilateral and bilateral resistance training significantly improved squat and rear-foot elevated split squat 1RM in rugby players, demonstrating the efficacy of both modalities for strength development. Despite similar strength outcomes, the underlying neuromuscular adaptations may diverge between approaches [23]. Compared to bilateral training, unilateral training imposes greater demands on neuromuscular coordination due to a narrower base of support, which necessitates increased joint stabilization and co-contraction [46]. McCurdy et al. [47] further demonstrated that, under equivalent relative loads, unilateral squats elicit greater electromyographic activity in the gluteus medius and hamstrings than bilateral squats, indicating enhanced hip stabilization and posterior chain engagement. However, this additional demand for stability may reduce the force output of the primary muscle groups and limit the ability to increase external load, thereby constraining further development of maximal strength [47]. It is worth noting that in our study, similar improvements in maximal strength were observed following both unilateral and bilateral training, which may be attributed to the matched training loads. This ensured comparable overall training intensity and volume between groups, leading to similar gains in strength.

According to the principle of training specificity, adaptations are optimized when the mechanical characteristics of training closely mirror those of the target movement, including contraction type, joint angles, and force–velocity profiles [21]. Unilateral training is generally more effective for enhancing single-leg performance, while bilateral training better supports tasks requiring synchronized bilateral force production [48]. Botton et al. [49] proposed that unilateral training may augment the bilateral deficit—evident in tasks like jumping and leg extension—by increasing neural drive during unilateral efforts without proportionately enhancing interlimb coordination. Empirical findings have also indicated that unilateral training is more effective for improving performance in unilateral-dominant activities such as sprinting, cutting, and single-leg jumping [50]. Bogdanis et al. [24] further found that unilateral plyometric training led to significantly greater improvements in single-leg countermovement jump (CMJ) performance and unilateral maximal isometric leg press strength compared to bilateral training, indicating a more targeted adaptation to unilateral motor tasks. This advantage may stem from the substantial increase in the rate of force development (RFD) induced by unilateral plyometric training, which enhances unilateral force output and explosiveness, further supporting the role of unilateral training in improving unilateral performance [24].

However, our findings contradict previous research supporting the training specificity principle, as we observed that unilateral training not only effectively improved strength but also facilitated its transfer to bilateral performance. This is in line with previous studies by Bogdanis et al. [24] and Makaruk et al. [27], who reported comparable improvements in bilateral jump performance and peak power following 6–12 weeks of unilateral and bilateral plyometric training. Although unilateral training may increase bilateral deficit, it does not necessarily inhibit bilateral performance gains. In fact, the greater stability demands and slower contraction speeds involved in unilateral movements may allow muscle groups to function closer to their optimal force-producing conditions, resulting in larger impulse generation and superior force output during unilateral actions [51,52]. Conversely, bilateral movements are often executed at higher contraction speeds, which may compromise maximal force production under equivalent loads. This could explain the lack of significant CMJ improvements in the BCCT group [51].

Furthermore, the neural adaptations induced by unilateral training are not entirely localized. Instead, they may involve whole-body neuromuscular coordination and, under certain conditions, contribute to improvements in bilateral performance—for example, through the cross-education effect [53]. Weir et al. [54] demonstrated that eight weeks of eccentric resistance training on the non-dominant leg yielded strength increases in both limbs, including improvements in bilateral isometric and eccentric strength. Notably, unilateral training induced more pronounced gains in both trained and untrained limbs compared to bilateral training. A similar cross-education effect may have contributed to the UCCT group's improvements in bilateral performance observed in our study.

In addition, Makaruk et al. [27] highlighted distinct temporal patterns in the performance effects of unilateral and bilateral plyometric training. Due to their higher neuromuscular coordination demands, unilateral protocols tend to induce rapid, short-term performance gains, which may decline quickly post-intervention due to limited external loading capacity. In contrast, bilateral training provides slower but more sustainable progress, with better retention of strength and power

following cessation. These findings suggest that unilateral training is particularly beneficial for short-term peaking of explosive performance, whereas bilateral training is more suitable for long-term development and maintenance of muscular strength.To maximize neuromuscular adaptations, previous studies have commonly recommended combining unilateral and bilateral training within the training process [48]. This approach allows for both short- and long-term training benefits while helping to avoid adaptation plateaus or performance decline caused by relying on a single training modality.

## 5. Conclusion

Both unilateral and bilateral complex-contrast training improved lower limb maximal strength and horizontal jump performance in female collegiate volleyball players. Unilateral training led to greater gains in vertical jump ability compared to bilateral training.

## 6. Limitations and future directions

A key limitation of this study lies in the difficulty of accurately matching total training volume between the UCCT and BCCT protocols. Although both groups trained at 85% of their 1RM with identical sets and repetitions, the inherent biomechanical differences between unilateral and bilateral exercises likely resulted in discrepancies in actual workload. For instance, research indicates that approximately 85% of the external load in Bulgarian split squats is borne by the front leg [47], while bilateral exercises have shown asymmetries in net joint torque between the left and right limbs [55]. These biomechanical disparities make it inherently challenging to equate total work output across training modalities [50]. Moreover, the 8-week intervention did not incorporate progressive increases in load, volume, or movement complexity.

While this helped maintain consistency in program delivery, it may have constrained the extent of neuromuscular adaptations in both groups. Another limitation is the relatively small sample size (n = 16), which may have reduced statistical power and increased the risk of Type II error, thereby limiting the generalizability of the results. Finally, the absence of a passive control group receiving only standard volleyball training limits the ability to isolate the intervention effects from natural training adaptations or external confounders. Future studies should consider larger sample sizes, more refined methods for equating workload across training types, and the inclusion of non-intervention control groups to enhance causal inference and clarify the sport-specific applicability of unilateral and bilateral CCT approaches.

## 7. Practical application

This study is the first to compare the effects of UCCT and BCCT on the strength and jump performance of female college volleyball players. The results show that both UCCT and BCCT significantly enhance vertical and horizontal jump abilities as well as maximal strength. This study indicates that both UCCT and BCCT effectively improve the vertical and horizontal jump abilities and maximal strength of female volleyball players. However, UCCT demonstrates a potential advantage in improving vertical jump performance.

Based on these findings, it is recommended that coaches incorporate UCCT into their training programs, especially for volleyball movements that require rapid initiation, precise control, and unilateral strength balance, such as blocking, serving, and spiking. Additionally, UCCT enhances neuromuscular coordination and strength balance, which helps improve athletes' overall performance and reduce the risk of injury. Therefore, UCCT can serve as an effective complement to traditional BCCT, further enhancing athletes' lower limb strength and explosiveness.

## Supporting information

**S1 File. Study data anonymized.**
(XLSX)

**S1 Fig. Graphical abstract.**
(TIF)

## Acknowledgments

We would like to thank the researchers and study participants for their contributions.

## Author contributions

**Conceptualization:** Ruixiang Yan.

**Data curation:** beiwang Deng, Ruixiang Yan.

**Formal analysis:** beiwang Deng, Ruixiang Yan, Xinke Sui, Shicong Ding, Duanying Li.

**Funding acquisition:** Jian Sun.

**Investigation:** Gesheng Lin, Shicong Ding, Duanying Li, Jian Sun.

**Methodology:** beiwang Deng, Ruixiang Yan, Shicong Ding, Duanying Li, Jian Sun.

**Project administration:** Jian Sun.

**Resources:** Shicong Ding, Jian Sun.

**Software:** Gesheng Lin.

**Supervision:** beiwang Deng, Ruixiang Yan, Xinke Sui.

**Validation:** Xinke Sui, Gesheng Lin, Shicong Ding, Duanying Li.

**Visualization:** Xinke Sui.

**Writing – original draft:** beiwang Deng.

**Writing – review & editing:** Duanying Li, Jian Sun.

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
