## [Decision Letter · Decision Letter 0]

22 Apr 2025

Dear Dr. Deng,

Thank you for submitting your manuscript to PLOS ONE. After careful consideration, we feel that it has merit but does not fully meet PLOS ONE’s publication criteria as it currently stands. Therefore, we invite you to submit a revised version of the manuscript that addresses the points raised during the review process.

We look forward to receiving your revised manuscript.

Kind regards,

Holakoo Mohsenifar

Academic Editor

PLOS ONE

 [Guangdong Provincial Philosophy and Social Sciences Regularization Project 2022 (GD22CTY09): Research on the Coordinated Development Path of International Competitiveness in Sports in the Guangdong-Hong Kong-Macao Greater Bay Area.]. 

4.  Please include a caption for figure 2.

5. Please include a copy of Table 1 and 2 which you refer to in your text on page 8.

6. We notice that your tables are uploaded with the file type ' Supporting Information'. Please amend the file type to 'Table’.

Reviewers' comments:

Reviewer's Responses to Questions

**Comments to the Author**

1. Is the manuscript technically sound, and do the data support the conclusions?

Reviewer #1: No

Reviewer #2: Partly

Reviewer #3: Partly

2. Has the statistical analysis been performed appropriately and rigorously?

Reviewer #1: No

Reviewer #2: Yes

Reviewer #3: Yes

3. Have the authors made all data underlying the findings in their manuscript fully available?

Reviewer #1: No

Reviewer #2: No

Reviewer #3: Yes

4. Is the manuscript presented in an intelligible fashion and written in standard English?

Reviewer #1: Yes

Reviewer #2: Yes

Reviewer #3: Yes

Reviewer #1: As a reviewer for PLOS ONE, I would like to acknowledge the authors for their efforts in conducting this study on the comparative effects of unilateral and bilateral complex contrast training on lower limb strength and jump performance in collegiate female volleyball players. The topic is relevant to sports science and contributes to the understanding of training methods for athletic performance enhancement. However, while the study presents valuable insights, the statistical analysis and tables lack clarity, making it difficult to fully interpret the findings. To ensure transparency and reproducibility, I recommend a thorough revision of the results section, including clearer statistical reporting, effect sizes, confidence intervals, and better-structured tables. Additionally, enhancing the discussion with a more critical analysis of the findings in relation to existing literature would strengthen the paper. If these revisions are addressed, I believe the article has the potential to meet the journal's standards for publication.

Title: change the title or delete “Experimental Study on”

Abstract:

L15-18: rewrite this sentence and avoid to used upper capital in the beginning of some words

L18: “A total of 16 female volleyball players were randomly assigned to either the UCCT group (n=8) or the BCCT group (n=8)”; randomly by what ??? player position, biologic age or what ????

Add more information about participants

L20-23: rewrite this sentence or delete “to assess changes in lower limb strength, vertical jump ability, and horizontal jump ability”

Add sentence about statistical tool used in this study

The result and conclusion are not clear, please rewrite

Keywords: Changed keywords (avoid used keywords used in the title)

Introduction:

The introduction of the article "Experimental Study on the Comparative Effects of Unilateral and Bilateral Complex Contrast Training on Lower Limb Strength and Jump Performance in Collegiate Female Volleyball Players" lacks clarity in establishing the research gap and justification for the study. While it provides a general overview of complex contrast training (CCT) and its potential benefits, it fails to offer a strong theoretical foundation explaining why unilateral versus bilateral training may yield different outcomes. Additionally, the introduction does not sufficiently engage with existing literature to highlight inconsistencies or unresolved questions, making it difficult to understand the novelty of the study. A more structured argument, supported by a critical review of past research, would strengthen the rationale for the investigation and improve its overall impact. REWRITE

Methods:

L76-77: rewrite this sentence

L80: “18 participants were randomly assigned to either the UCCT group (n=9) or the BCCT group (n=9) using a random number-based randomization method” HOW ??? more information

L85: changed “Squat 1RM” by 1-RM squat and in all the text

Table 1 ?????? where is

Table 2 ?????? where is

Table 3: add abbreviation list

L188-189: Tukey post hoc, in this study you don’t need post hoc because you have 2 groups and 2 times

Result:

The results and tables presented in this article exhibit several shortcomings that affect their clarity and interpretability. While the tables provide numerical data on performance changes, they lack sufficient statistical indicators, such as effect sizes or confidence intervals, to contextualize the significance of the findings. Furthermore, the presentation of results does not clearly differentiate between practical and statistically significant outcomes, making it difficult to determine the real-world applicability of the findings. The authors also fail to discuss potential variability in individual responses, which is crucial in sports science research. A more detailed analysis, including graphical representations and a clearer explanation of key trends, would improve the comprehensibility and impact of the results section.

Discussion:

The discussion section of the article "Experimental Study on the Comparative Effects of Unilateral and Bilateral Complex Contrast Training on Lower Limb Strength and Jump Performance in Collegiate Female Volleyball Players" is weakened by the lack of clarity in the results and tables presented. Because the results are not well-structured or sufficiently detailed, the discussion fails to provide a compelling interpretation of the findings. The authors attempt to compare their results with previous studies, but without clear statistical analysis or well-presented data, these comparisons remain vague and inconclusive. Additionally, the discussion lacks a critical evaluation of potential limitations. Without a more precise and structured presentation of results, the discussion becomes speculative rather than evidence-based, making it difficult to draw meaningful conclusions or practical applications for training protocols. A more rigorous analysis and transparent data reporting are necessary to strengthen the discussion and support the study’s claims.

Reviewer #2: Thank you for the opportunity to review this manuscript. In general, I believe the introduction needs a deeper discussion of more relevant topics to set the scene for the study, and the discussion needs to be much more balanced. I have made specific comments below.

Abstract:

Line 22 – suggest writing 1RM in full

Line 25 – would be good to state the direction of the group X time effect here to highlight ¬which intervention was more effective.

Introduction:

Line 43 – suggest also adding some commentary around power/plyometric training here as a another potential method of improving performance. This will lead into the next paragraph where you then talk about how they can be combined using CCT.

Line 44 to 48 – this section is quite brief. It would be good to describe some applied research examining similar outcomes using CCT. Schneiker et al., 2023 and Luders et al., 2024 are two recent studies that investigated the topic that could be discussed.

Line 46 – can you provide an example of what this might look like? (i.e., back squat into box jumps, or something like that)

Line 47 – need to expand on PAPE in more detail as it is what underpins the potential benefit of CCT.

Line 49 – this sentence doesn’t really suggest that CCT is worth investigating (i.e., why not just do plyometric exercises then strength exercises?). Suggest using the papers mentioned above to build a stronger rationale for CCT in your population group.

Line 56 to 63 – this section uses very broad language, and the point you make isn’t particularly clear. It would be better to refer to some literature looking at applied outcomes in response to unilateral and bilateral strength training as a way to highlight what the benefits of both are. The meta-analysis by Zhang et al., 2023 on the topic would be a good starting point.

Line 64 to 71 – would be good to mention in here that (at least to my knowledge) no study has compared unilateral to bilateral CCT, which makes this a rather novel project.

Line 71 – this hypothesis is different to what you started in your pre-registration, whereby you said you were exploring the two modalities to simply see what was better. This hypothesis/aim should be the same as your pre-registration.

Methods:

Line 79 – can you also describe where in the competitive season this was? Was it pre-season, in season, etc. Moreover, can you speak to their training in the prior months before the study. Were they regularly conducting resistance training?

Line 81 – can you elaborate on the randomisation method? Was this an online tool?

Line 91 – need to provide more information on why you thought an effect size of two very similar interventions would be medium? This seems unrealistic to expect.

Line 93 – was this a convenience sample? If so, this needs to be stated.

Line 93 – can you provide more information on the level of the athletes? What level of competition were they competing in?

Line 103 – can you speak to the experience of the athletes with these exercises specifically? Had they all been doing squats and Bulgarian split squats in their training, or were they all new exercises?

Line 104 – can you add the range of attendance % for both groups in brackets here as well.

Line 120 – who provided/supervised the intervention? Or was it conducted alone?

Line 125 – can you also provide the rest between the conditioning activity and the plyometric activity? And in this section it would be good to outline why you chose different plyometric exercises for each group.

Line 129 – can you also list the days for training and games, as you did above with the gym sessions.

Line 138 to 146 – can you provide a bit more information here on the instructions provided to participants, and whether verbal encouragement was used.

Line 150 – would be good to provide some additional context as to what the purpose of this measure is.

Line 152 – as above, could you provide the verbal instructions that were given to participants? And were there toes or heels placed on the starting line?

Line 159 – in the training program you imply that the UCCT group uses 85% of their Bulgarian split squat 1RM. Was this measured? And if so, how?

Line 170 – what specific measures/data were used to calculate these?

Results:

General comment – throughout the results it would be good to highlight the direction of changes. For example, both groups improved XX, UCCT saw larger improvements in XX, etc.

Table 4 – The post-test column for the BCT group looks like it has incorrect data? It would be good to present the mean change and 95% CI for the mean change for both groups here somewhere. Could also do the between group mean difference and 95% CI's.

Discussion:

General comment: I believe the discussion should be changed significantly. It currently reads like an advertisement for single leg training, rather than providing an accurate and balanced discussion of the literature. I would suggest tempering your language significantly throughout, and just focusing on the results of your study rather than looking to adjacent areas of research to highlight the benefits of unilateral CCT. I have also made some more minor comments below, that may not be relevant once you make larger changes.

Line 227 – this first sentence is out of place here.

Line 228 to 234 – similar to my general comment in the results, you need to be more descriptive. And there is no need to repeat effect sizes etc. here.

Line 235 – dio both of these studies use CCT? It looks like one was just RT?

Line 241 – the review you refer to doesn’t show this? Could look for more relevant citations or be more speculative in your commentary (as this is a speculative statement)

Line 243 – this study looked at running, not unilateral resistance training. Remove this sentence.

Line 246 – this citation shows bilateral training results in more quadriceps activity than unilateral training, so this doesn’t support this statement. Suggest editing line 246 to 249 to more accurately reflect the research and be more cautious in your interpretation of your results.

Line 261 – you have a very small sample size that could have explained this difference in groups. Suggest removing this sentence as the finding was not significant.

Line 285 – citation 60 does not support this statement. Suggest removing.

Line 296 – was Bulgarian split squat 1RM measured? If so, it should be discussed?

Line 316 to 335 – can remove this paragraph. The references are not relevant to your study, and the role of transient acute increases in testosterone effective chronic training adaptations has been debunked.

Line 344 to 356 – this can also be removed. It reads like an advertisement for single leg training, rather than a balanced evaluation of the literature.

Limitations:

Also need to comment on the low sample size, and the fact that these findings may not generalise beyond this population. And that you did not compare this intervention to a “traditional” training program. As such, it is unclear whether these results are due to CCT or due to the unilateral exercises. Also, if you did not assess Bulgarian split squat 1Rm that should be noted as a limitation, as how do you know they were at 85% of their 1RM?

Reviewer #3: This study aimed to compare the effects of unilateral and bilateral complex training on lower limb strength and jump ability in female college volleyball athletes. While the paper is well-discussed, there are several points that require clarification or editing for improved clarity.

L81: Are 9 participants per group enough for your study’s randomized parallel-controlled trial design? A larger sample size would strengthen the validity of the study results and provide more robust conclusions.

L91: Given that the power was set at 0.5 in the G-power analysis, it is essential to consider if this value was based on previous research in the field. Can the authors provide a relevant citation that justifies the choice of this power size for determining the sample size in the study? Clarifying the rationale behind selecting this specific power level would strengthen the validity of the sample size calculation.

The study's limitation of a small sample size should be acknowledged, especially considering that only 8 athletes in each group completed both the training and assessments. This information highlights the importance of interpreting the results with caution due to the relatively limited number of participants who fully participated in the study.

Table 1.: Please change “Weight” with “Body mass”

Table 2: What is the number of groups? Do you mean number of sets?

L121-125: A significant limitation of the study is the lack of volume matching between the two training protocols (UCCT and BCCT). Without equating the total training volume between the experimental groups, it becomes challenging to attribute any observed differences in outcomes solely to the type of training.

Both resistance and plyometric training exercises should be equated in volume.

Resistance: number of sets × total number of repetitions × load

Plyometric: number of foot contacts

According to Table 2, the training intervention lacks progressive overload, as indicated by the absence of intensity, volume, or complexity adjustments over the 8-week training period. Without systematically increasing the training stimulus to challenge the athletes' physiological capacities, the potential for optimal improvements in explosive power may be limited.

128-129: Were all the athletes from the same volleyball team? Did they participate in the same volleyball training sessions?

L139-146: Please describe also how the participants landed.

Table 4: Mean values for BCCT are missing; there is only SD.

Table 4: Please change all instances of "BCT" to "BCCT" and "UCT" to "UCCT" for consistency.

L270: Please add bibliography.

L362: Authors should include in the study's limitations the absence of a control group that could have engaged solely in the volleyball training sessions.

**Do you want your identity to be public for this peer review?** For information about this choice, including consent withdrawal, please see our Privacy Policy

Reviewer #1: **Yes: ** Mehrez Hammami

Reviewer #2: No

Reviewer #3: No

---

## [Author Response · Author response to Decision Letter 1]

8 May 2025

Dear Academic Reviewers,

We would like to express our heartfelt thanks to you for your thoughtful, constructive, and highly valuable comments on our manuscript.

Your suggestions have provided us with important insights and have significantly contributed to improving the quality, clarity, and scientific rigor of our work. We truly appreciate the time, expertise, and effort you have dedicated to reviewing our manuscript.

In this revision, we have carefully considered every point raised, and we have revised the manuscript accordingly. Detailed, point-by-point responses are provided in the attached document titled “Response to Reviewers.docx.”

We remain deeply grateful for your guidance and support throughout the review process.

Sincerely,

Beiwang Deng

---

## [Decision Letter · Decision Letter 1]

1 June 2025

Thank you for submitting your manuscript to PLOS ONE. After careful consideration, we feel that it has merit but does not fully meet PLOS ONE’s publication criteria as it currently stands. Therefore, we invite you to submit a revised version of the manuscript that addresses the points raised during the review process.

We look forward to receiving your revised manuscript.

Kind regards,

Holakoo Mohsenifar

Academic Editor

PLOS ONE

Journal Requirements:

Reviewers' comments:

Reviewer's Responses to Questions

**Comments to the Author**

Reviewer #2: (No Response)

Reviewer #3: All comments have been addressed

2. Is the manuscript technically sound, and do the data support the conclusions?

Reviewer #2: Partly

Reviewer #3: Yes

3. Has the statistical analysis been performed appropriately and rigorously?

Reviewer #2: Yes

Reviewer #3: Yes

4. Have the authors made all data underlying the findings in their manuscript fully available?

Reviewer #2: Yes

Reviewer #3: Yes

5. Is the manuscript presented in an intelligible fashion and written in standard English?

Reviewer #2: Yes

Reviewer #3: Yes

Reviewer #2: Thank you for addressing most of my prior comments. I have some additional (and some repeated) comments that I believe need to be addressed further. Please note that these comments relevant to non-tracked changes version of manuscript.

Abstract:

Line 21 – spss needs to be written in full. Although you could remove “in SPSS software” from the abstract as this information is in the methods.

Line 24 – 1-RM needs to be written in full before using the abbreviation.

Introduction:

Line 40 – suggest removing the word “level” from “vertical jump level”

Line 78 – you should note that many of the studies included in this review don’t use comparative interventions (i.e., some compare CCT to RT alone or plyometric activity alone), which makes it harder to determine its effectiveness. This is where you could also discuss Luders et al., 2024 and Schneiker et al., 2023 in a little more detail, as they are two studies that used a volume and intensity matched intervention like yours (albeit with a bilateral focus), rather than simply citing them above.

Line 92 – this sentence does not really add anything meaningful and is really just a bunch of buzzwords. Suggest removing.

Line 109 – in line with my earlier comments, I would suggest removing the hypothesis entirely as based upon your pre-registration it is clear this is not the hypothesis you had when designing the study. This looks like a clear example of HARKing.

Methods:

Line 128 – write SPSS in full

Line 131 – need to write all these abbreviations in full the first time they appear in the manuscript.

Line 141 – It seems like you didn’t really think the sample size through here OR tried to find an effect size that firs your sample size. You used a study that added a power focused training program to normal skills training, and then compared it to a control group that only did normal skills training. This effect size is likely to be much larger than what you would expect to see when comparing two similar interventions (like your study). In future, suggest either being more thoughtful when you conduct your power calculation OR state that you used a convenience sample if that is what you did.

Line 158 – body mass doesn’t need to be capitalized.

Table 3 – suggest noting that the rest periods were in minutes in this table

Line 211 – suggest changing “better” to “best”

Line 232 – suggest noting this was back squat and Bulgarian squat here

Results:

Line 284 to 288 – I would suggest removing the commentary around larger effect sizes indicating better results here. The fact that there was no group x time interactions present between these variables would suggest that these differences are potentially due to chance (indeed, if you reported the 95% CI for these effect sizes we would expect to see some notable variation.

Line 297 – can remove this sentence.

Line 302 – In figure 4 it looks like not all participants completed 1RM squat assessment. Can you please note why this is the case and report the numbers that did complete in text here.

Line 302 – should also add 1RM Bulgarian split squat to the results assuming you tested it at the end of the intervention? Can also add it to table 5.

Discussion:

Line 318 to 321 – in line with one of my earlier comments, you can remove this as it may have been to chance.

Conclusions:

Line 383 – suggest removing this sentence, as you pre-registration didn’t have any hypotheses presented.

Reviewer #3: (No Response)

**Do you want your identity to be public for this peer review?** For information about this choice, including consent withdrawal, please see our Privacy Policy

Reviewer #2: No

Reviewer #3: No

---

## [Author Response · Author response to Decision Letter 2]

4 Jun 2025

Dear Reviewers,

Thank you sincerely for your thoughtful and constructive feedback. Your careful evaluation and detailed suggestions have been invaluable in improving the clarity and rigor of our manuscript. We deeply appreciate the time and expertise you have generously invested in reviewing our work. All comments from Reviewer #2 have been addressed in the revised manuscript, and changes are highlighted in red. Reviewers #1 and #3 did not provide additional comments, but we remain grateful for their oversight. Your contributions are essential to elevating the quality of our research, and we are truly grateful for your guidance.

Respectfully,

Beiwang Deng

---

## [Editor Report · Decision Letter 2]

12 June 2025

Effects of Unilateral and Bilateral Complex-Contrast Training on Lower Limb Strength and Jump Performance in Collegiate Female Volleyball Players

PONE-D-25-07993R2

Dear Dr. Jian Sun,

We’re pleased to inform you that your manuscript has been judged scientifically suitable for publication and will be formally accepted for publication once it meets all outstanding technical requirements.

Kind regards,

Holakoo Mohsenifar

Academic Editor

PLOS ONE
---

## [Editor Report · Acceptance letter]

PONE-D-25-07993R2

PLOS ONE

Dear Dr. Sun,

I'm pleased to inform you that your manuscript has been deemed suitable for publication in PLOS ONE. Congratulations! Your manuscript is now being handed over to our production team.

Kind regards,

on behalf of

Dr. Holakoo Mohsenifar

Academic Editor

PLOS ONE